# c-Cbl Regulates Murine Subventricular Zone-Derived Neural Progenitor Cells in Dependence of the Epidermal Growth Factor Receptor

**DOI:** 10.3390/cells12192400

**Published:** 2023-10-03

**Authors:** Maximilian Vogt, Madhukrishna Kolothara Unnikrishnan, Nora Heinig, Ulrike Schumann, Mirko H. H. Schmidt, Kathrin Barth

**Affiliations:** Institute of Anatomy, Medical Faculty Carl Gustav Carus, Technische Universität Dresden School of Medicine, 013107 Dresden, Germany; maximilian.vogt1@tu-dresden.de (M.V.); madhukrishna.kolothara_unnikrishnan@tu-dresden.de (M.K.U.); nora.heinig@tu-dresden.de (N.H.); ulrike.schumann@tu-dresden.de (U.S.); mhhs@mailbox.tu-dresden.de (M.H.H.S.)

**Keywords:** adult neurogenesis, c-Cbl, EGFR, neural stem cells, SVZ, ubiquitin ligase, rodents

## Abstract

The localization, expression, and physiological role of regulatory proteins in the neurogenic niches of the brain is fundamental to our understanding of adult neurogenesis. This study explores the expression and role of the E3-ubiquitin ligase, c-Cbl, in neurogenesis within the subventricular zone (SVZ) of mice. In vitro neurosphere assays and *in vivo* analyses were performed in specific c-Cbl knock-out lines to unravel c-Cbl’s role in receptor tyrosine kinase signaling, including the epidermal growth factor receptor (EGFR) pathway. Our findings suggest that c-Cbl is significantly expressed within EGFR-expressing cells, playing a pivotal role in neural stem cell proliferation and differentiation. However, c-Cbl’s function extends beyond EGFR signaling, as its loss upon knock-out stimulated progenitor cell proliferation in neurosphere cultures. Yet, this effect was not detected in hippocampal progenitor cells, reflecting the lack of the EGFR in the hippocampus. *In vivo*, c-Cbl exerted only a minor proneurogenic influence with no measurable impact on the formation of adult-born neurons. In conclusion, c-Cbl regulates neural stem cells in the subventricular zone via the EGFR pathway but, likely, its loss is compensated by other signaling modules *in vivo*.

## 1. Introduction

Adult neurogenesis, the process of generating new neurons in adulthood, is a dynamic and complex process that occurs throughout life in specific neurogenic niches of the brain, including the subventricular zone (SVZ) of the lateral ventricles and the subgranular zone (SGZ) of the dentate gyrus (DG) in the hippocampus (HC). Interactions between intrinsic and extrinsic signals provided by cells within these neurogenic niches, as well as from distant sources, affect the fate of neural stem cells (NSCs) and neural progenitor cells (NPCs or transient amplifying precursors (TAPs)). Importantly, the dysregulation of adult neurogenesis is linked to various neurological and psychiatric disorders, including depression, addiction, and neurodegenerative diseases [1,2].

Adult neurogenesis in the murine SVZ plays an important role in brain function, including olfactory bulb (OB) neurogenesis, neurovascular coupling, and brain repair after injury [3,4]. An alteration in the number of TAPs in the SVZ is implicated in a range of neurological disorders, including Huntington’s disease [5], Parkinson’s disease [6], vascular dementia [7], glioblastoma multiforme [8], Alzheimer’s disease [9,10], and cerebrovascular damage [11]. Thus, understanding the mechanisms that influence proliferation of NSCs in the SVZ is crucial for the development of therapeutic strategies for these diseases.

Multiple signaling pathways and transcriptional regulators have been identified as key players in controlling adult neurogenesis in the SVZ. These include but are not restricted to Notch, Wnt, and receptor tyrosine kinase (RTK) signaling, such as the epidermal growth factor receptor (EGFR) pathways [12,13,14], as well as transcription factors such as Ascl1 or NeuroD1 [15,16]. In order to understand the role of these signaling pathways in adult neurogenesis, illumination of the proteins involved in their regulation is required. One such group of regulatory elements are Casitas B-lineage lymphoma (Cbl) proteins, which play a crucial role in regulating RTK signaling. RTKs are transmembrane proteins that transmit signals from extracellular stimuli to intracellular signaling pathways and are important for various cellular processes such as proliferation, differentiation, and survival [17]. In the case of EGFR, recognized for its proliferation signaling and critical role in pathooncology, stopping this receptor’s inappropriate functioning is clinically important. Here, the ubiquitination process mediated by Cbl proteins plays a pivotal role. Upon ligand binding to EGFR, Cbl proteins, particularly c-Cbl, are recruited to the activated receptor. c-Cbl acts as an E3 ubiquitin ligase, attaching ubiquitin molecules to the EGFR. This ubiquitination serves as a signal for the internalization of the receptor via clathrin-mediated endocytosis (CME) at low ligand concentrations, whereby the receptor signal is retained for longer and a large proportion of the receptors are recycled instead of degraded. As ligand concentrations increase, CME decreases and non-clathrin-mediated endocytosis (NCE) increases, leading to the routing of the ubiquitinated EGFR to lysosomes for degradation, effectively attenuating the EGFR signaling pathway [18,19]. Throughout this process, Cbls act as negative regulators of RTK signaling through ubiquitination, thus targeting RTKs for degradation by the proteasome, based on the threshold system of binding ligand concentration [20].

Three human isoforms of Cbl proteins have been identified: c-Cbl, Cbl-b, and Cbl-3. c-Cbl and Cbl-b are similar in structure and function and are expressed in various tissues, including the brain. Cbl-3 is a more recently identified shorter isoform and is mainly expressed in epithelial cells [21]. Cbl proteins have been shown to be involved in oncogenesis, as mutations in c-Cbl and Cbl-b have been identified in various cancer types [22,23]. Furthermore, c-Cbl binding and ubiquitination-dependent lysosomal degradation of membrane-associated Notch1 have been demonstrated [24]. The role of Cbl proteins in brain regeneration has been highlighted by the MiR-22-3p inhibitor in recovery from neural cord injury through the EGFR pathway [25]. However, little is currently known about the localization and function of Cbl proteins in the neurogenic niches of the brain. 

In this study, we investigate the potential role of c-Cbl in adult neurogenesis in the SVZ and the hippocampus (HC). We analyzed the expression of c-Cbl and evaluated the effects of c-Cbl cell-specific knock-outs in NSCs and TAPs on adult neurogenesis in mice. Our results provide new insights into the mechanisms underlying the regulation of adult neurogenesis in the SVZ and HC by E3 ligases.

## 2. Materials and Methods

### 2.1. Animals

Animal experiments were approved by the Landesdirektion Chemnitz, Saxony, Germany (permit number TVV56/2020, AZ: 25-5131/496/59). Experiments were conducted according to the German Animal Welfare Law and the Directive 2010/63/EU for the protection of animals used for scientific purposes and followed the ARRIVE and GV-SOLAS guidelines for research on animal behavior. Mice were housed in climate-controlled, pathogen-free conditions at the Experimental Center of the Medical Theoretical Center (MTZ, Medical Faculty Carl Gustav Carus, Dresden, Germany) on a 12 h light–dark cycle with free access to food and water.

### 2.2. Mouse Lines

#### 2.2.1. *c-Cbl^fl/fl^;Cbl-b^KO^;Cbl-3^KO^;Ascl1-CreERT2*

*c-Cbl^fl/fl^;Cbl-b^KO^;Cbl-3^KO^* mice were provided by Mayumi Naramura from Hamid Band’s group (Eppley Institute for Cancer Research, Omaha, USA). They originated from Josef Penninger’s group (HZI, Braunschweig, Germany) and Hua Gu’s group (IRCM, Montreal, QC, Canada). In these mice, exon 4 of the *c-Cbl* gene was flanked by loxP sites [26]. A neomycin resistance cassette was inserted into exon 3 of the *Cbl-b* gene to achieve constitutive inhibition of Cbl-b expression [27]. For Cbl-3 deletion, a lacZ/resistance cassette was inserted in place of exon 1 to achieve constitutive inhibition of expression [28]. In *Ascl1-CreERT2* mice [29], the entire coding region of the endogenous achaete-scute complex homolog 1 (Ascl1) was replaced with a CreERT2 fusion protein. *c-Cbl^fl/fl^;Cbl-b^KO^;Cbl-3^KO^* mice crossed with Ascl1-CreERT2 mice thus allowed for the deletion of c-Cbl in proliferative TAPs upon the application of tamoxifen on the background of a constitutive Cbl-b and Cbl-3 double knock-out. *Ascl1-CreERT2* mice (JAX stock #012882) were obtained from the Jackson Laboratory (Bar Habor, ME, USA).

#### 2.2.2. *c-Cbl^fl/fl^;Cbl-b^KO^;Cbl-3^KO^;GLAST-CreERT2*

*c-cbl^fl/fl^;cbl-b^KO^;cbl-3^KO^* mice described above were crossed with *GLAST-CreERT2* animals [30], which express a CreERT2 fusion protein under the control of the astrocyte-specific glutamate–aspartate transporter (GLAST) promoter for the tamoxifen-induced deletion of c-Cbl in astrocytes and NSCs. Additionally, Cbl-b and Cbl-3 were constitutively deleted in these mice as described above. *GLAST-CreERT2* mice were provided by Magdalena Götz (Institute for Stem Cell Research, Helmholtz Zentrum, München, Germany).

### 2.3. Mouse Treatments

**BrdU, CldU**: a 10 mg/mL solution of BrdU or CldU was prepared and 100 µL of solution was intraperitoneally (ip) injected. **IdU**: 500 µL of a 2 mg/mL solution was ip injected. Hence, mice received about 1 mg base analogue per animal (approximately 40 mg/kg). **Tamoxifen**: mice received daily injections of tamoxifen in peanut oil/ethanol (9:1) as a vehicle ip at a dose of 3 mg per 20 g body weight for 3 consecutive days.

### 2.4. Tissue Preparation

Mice were anesthetized by the administration of a ketamine–xylazine mixture ip (100 μL per 10 g body weight, 450 mg/kg ketamine and 48 mg/kg xylazine, both in a 0.9% sodium chloride solution). Subsequently, mice were perfused transcardially with phosphate-buffered saline (PBS), followed by 4% paraformaldehyde in PBS. The high dose of ketamine–xylazine represents an overdose to kill the animals by perfusion.

Brains were removed and fixed in 4% paraformaldehyde in PBS at 4 °C for 24 h. Subsequently, brains were successively transferred into 15% and 30% sucrose solution in PBS for cryopreservation. Brains were cut on a cryostat (CM1950 from Leica) to obtain 20 µm thick sections which were stored in cryoprotective solution at −20 °C. Alternatively, brains were embedded in Tissue-Tek O.C.T. solution (Sakura Fintek, Torrance, CA, USA), frozen on dry ice, and stored at −80 °C until further processing. 

### 2.5. Immunofluorescence (IF) Staining

After application to slides, the sections were dried for 1 h at room temperature (RT). In the case of staining against nuclear epitopes, specimens were boiled for 30 min in preheated citrate buffer, then cooled and washed several times in PBS. Sections were incubated for at least 1 h at RT in blocking solution (5% BSA, 0.1% Triton in PBS) and overnight at 4 °C with primary antibody solution (in 1% BSA, 0.1% Triton in PBS, dilution of primary antibody shown in Table 1). After multiple washing steps using PBS solution, the specific staining of the antibodies was counterstained with species-specific fluorophore-conjugated secondary antibodies at RT for 1 h under light protection and the fluorescent dye 4′,6-diamidino-2-phenylindole (DAPI) to stain cell nuclei. Sections were washed with distilled water and mounted in Fluoromount (Thermoscientific, Waltham, MA, USA). Images were acquired using a transmitted light microscope (AxioScanZ.1 from ZEISS of CFCI, MTZ, Dresden, Germany) with a 20×/0.8 NA objective and Apotome II attached. 

### 2.6. Immunoperoxidase (IP) Staining

If necessary, sections were deparaffinized and then washed in xylene, followed by a series of decreasing concentrations of ethanol and H_2_O three times for 2 min each. Subsequently, heat epitope retrieval was performed by boiling the sections in citrate buffer for 5 min in a microwave at 350 W. After the sections had cooled down, they were washed twice in PBS buffer for 5 min each and were incubated in PBS with 0.2% H_2_O_2_ for 30 min to reduce endogenous peroxidase activity under gentle stirring. The sections were then blocked with a solution containing 2% serum and 1% Triton in PBS for 20 min. After blotting off the blocking solution, the sections were incubated overnight at 4 °C with the primary antibody being diluted in PBS buffer (dilutions shown in Table 1). After washing in PBS, the secondary antibody was diluted in PBS buffer containing 1% serum solution and was incubated with the sections for 45 min at RT. After another washing step in PBS, sections were biotinylated with an avidin–biotin complex (ABC) solution for 45 min and washed again in PBS. Under strict microscopic control, 3,3’-diaminobenzidine (DAB) solution with activated H_2_O_2_ was incubated with the sections for 5 min and the reaction was terminated by rinsing and washing in H_2_O. Eventually, sections were stained with hematoxylin for 30 s and rinsed in tap water for 10 min. Finally, the slides were dehydrated through a series of increasing concentrations of ethanol followed by xylene solutions and were mounted in Depex.

### 2.7. RNAscope

RNAscope is commercially available from Advanced Cell Diagnostics (ACD, Newark, CA, USA) [31]. ACD has a large library of validated probes for mouse genes. Typically, the standard probes contain 20 ZZ probe pairs (50 base pairs (bp)/pair) covering 1000 bp, but the probes can be longer or shorter depending on the desired target region. Here, we used a probe against mouse c-Cbl (Mm-Cbl-C1, NM_007619.2) as well as probes for positive (SR-RNU6-S1) and negative (SR-Scramble-S1) controls. The protocol was applied according to the manufacturer’s guidelines for 20 μm sections (standard protocol from ACD: RNAscope™ Multiplex Fluorescent Reagent Kit v2). 

### 2.8. Flow Cytometry

Cell populations were identified by flow cytometry as previously described [32]. Wild-type mice were sacrificed by cervical dislocation and the SVZ and HC were microdissected. Three mice per sample were pooled followed by digestion and dissociation using the Neural Tissue Dissociation Kit (T) (Miltenyi Biotec, Bergisch Gladbach, Germany) according to the manufacturer’s instructions. A BD FACSAria II device (BD Biosciences, Franklin Lakes, NJ, USA) was used to purify NSCs, TAPs, and astrocytes by fluorescence-activated cell sorting. The cells were incubated for 30 min at 4 °C in the dark in PBS with the following antibodies and conjugates: PE-conjugated anti-mGLAST (1:50, clone ACSA-1, Miltenyi Biotec), PE-Cy7-conjugated anti-mCD133 (1:100, clone 315-2C11, Biolegend, San Diego, CA, USA), Alexa FluorTM 488-complexed EGF (1:100, Invitrogen, Carlsbad, CA, USA). For negative selection, the following antibodies were used: APC-Cy7-conjugated anti-mCD45 (1:200, Becton Dickinson, Franklin Lakes, NJ, USA) and APC-Cy7-conjugated anti-mTer119 (1:100, Biolegend). Cells from the HC and the SVZ zone were selected using a combination of the following markers: GLAST-positive^(+)^/CD133^+^ for NSCs, GLAST-negative^(−)^/CD133^−^/EGFR^+^ for TAPs, and GLAST^+^/CD133^−^ for astrocytes.

### 2.9. RNA Extraction 

RNA extraction was performed using tissue and sorted cells from the mouse SVZ. Mouse brains were mechanically disrupted using a mortar and pestle precooled in liquid nitrogen. Both tissue and sorted cells were lysed in RLT buffer (Qiagen, Germantown, Maryland, US). RNA was purified and gDNA was removed using an RNeasy Plus Mini Kit (Qiagen) according to the manufacturer’s instructions. 

### 2.10. Quantitative Reverse Transcriptase Polymerase Chain Reaction (qRT-PCR)

cDNA was synthesized from isolated RNA using the Revert Aid H Minus First Strand cDNA Synthesis Kit (Thermoscientific, Waltham, MA, USA) following the manufacturer’s specifications. qRT-PCR was performed using SsoAdvanced Universal IT Sybr Green Set (Thermoscientific, Waltham, MA, USA). PCR and amplicon detection were performed by a real-time PCR system (CFX96 RT-PCR System, Bio-Rad, Hercules, CA, USA). For quantification, expression levels were normalized against the housekeeping genes glycerinaldehyde-3-phosphate dehydrogenase (GAPDH) and ribosomal protein 13 (RPS13). To analyze quantitative expression levels among groups, the 2(-Delta Delta C(T)) method was applied [33]. Primer sequences are listed in Table 2.

### 2.11. Neurosphere Assays

Primary neurosphere cultures were established for 3 control (Cre^−^) and 3 knock-out (Cre^+^) animals of the *c-Cbl^fl/fl^;Cbl-b^KO^;Cbl-3^KO^;Ascl1-CreERT2* line (8–12 weeks old). Mice were killed by cervical dislocation and brains were removed. SVZ and HC were dissected following the protocol of Walker and Kempermann [34]. Sections were enzymatically dissociated for 30 min in Leibowitz L-15 medium, containing 0.8 mg/mL papain and 0.5 mM EDTA (at 37 °C), with mechanical dissociation every 5 min by shaking and a fire-polished pipette. Further, sections were centrifuged at 300 g for 5 min and the supernatant was removed. The cell pellet was resuspended with 100 μL of PBS buffer and the volume topped up to 500 μL. After centrifugation for 3 min at 300 g, the supernatant was cleanly removed and the cell pellet was carefully resuspended in 1 mL of culture medium. Finally, the cell suspension was dissolved in 9 mL of culture medium (DMEM/F-12 medium containing 1mM HEPES, 10 mg/mL penicillin, 10 mg/mL streptomycin, 1B27 supplement, and 20 ng/mL EGF plus FGF (Thermoscientific, Waltham, MA, USA)). The resulting solution was divided onto 24-well plates at 1 mL per well and incubated at 37 °C and 5% CO_2_. Neurospheres of the SVZ were microscopically examined after 7 d (spheres of the HC after 12 d) using an AxioZoom.V16 from Zeiss (Core Facility Cellular Imaging, Medizinisch-theoretisches Zentrum, Dresden). The diameters of all neurospheres were determined using QuPath [35] and were statistically analyzed by 2-way ANOVA in GraphPad Prism 8.0 software (Statcon, Witzenhausen, Germany).

### 2.12. Statistical Analysis of Anatomical and Biological Data

The number of positive cells was determined in the first three cell layers of the lateral ventricular wall adjacent to the ependyma and plotted against the length of the ventricle (cells^+^/mm). Positive cells in the OB were determined in both the stratum granulosum and the stratum mitrale and plotted against the analyzed area per animal (cells^+^/mm^2^). Data are presented in figures as mean ± standard deviation (SD). Statistical analyses were performed using GraphPad Prism 8.0 software. Student’s *t*-test or Mann–Whitney *U* test was used for statistical analysis of three or more biological replicates per experiment and a *p* value smaller than 0.05 (*p* < 0.05) was considered significant. All experiments and analyses were carried out without knowledge of the genotype or treatment group. Parametric data were analyzed using an unpaired two-tailed Student’s *t*-test. Non-parametric data or small sample sizes were analyzed with the Mann–Whitney *U* test. Statistical methods are described in figure legends or specific experimental procedures.

## 3. Results

### 3.1. c-Cbl Is Localized in the SVZ and Expressed in NSCs and TAPs

Adult NSCs in the SVZ co-exist in a quiescent (qNSC) or an actively dividing (aNSC) state [36]. Upon activation, they give rise to TAPs, which in turn generate neuroblasts (NBs) that migrate to the OB and differentiate into interneurons [37].

In order to investigate the localization and expression of c-Cbl in the SVZ, in particular in NSCs (B cells) and TAPs (C cells), murine specimens were analyzed by IP and IF staining. IP staining in the lateral wall of the lateral ventricle revealed a weak signal of c-Cbl in the SVZ (Figure 1A,B). Mouse kidney specimens served as a positive control, with c-Cbl expression being spotted in the cytoplasm of glomerular podocytes, in the collecting duct epithelia, and in endothelia (Appendix A; [38]; https://www.proteinatlas.org/ENSG00000110395-CBL/tissue/kidney, accessed on 20 June 2023, 16:34). Further, the expression of murine c-Cbl in mouse specimens was analyzed by RNAscope to study the c-Cbl expression pattern in the neurogenic niche in greater detail. However, the SVZ displayed only a few c-Cbl-expressing cells (Figure 1C).

To identify the cellular origins of c-Cbl, SVZ tissue was dissociated and the cells were separated by fluorescence-activated cell sorting using three cellular markers (GLAST^−^ or ^+^/CD133^−^ or ^+^/EGFR^−^ or ^+^) to discriminate between NSCs, TAPs, and astrocytes [32] (Appendix A). Sorted cells were subjected to qRT-PCR analyses, which revealed a ubiquitous transcription of c-Cbl with only small differences in transcript abundance among cell types. c-Cbl was synthesized almost equally as compared with unsorted SVZ tissues for NSCs (1.25 ± 0.25-fold), TAPs (0.67 ± 0.19-fold), and astrocytes (1 ± 0.16-fold) in the cell types examined (Figure 1D).

Subsequently, c-Cbl transcription was visualized in situ by a combination of c-Cbl-specific RNA probes (RNAscope) and cell-specific IF markers. More than 90% of the glial fibrillary acidic protein (GFAP)^+^/GLAST^+^/CD133^+^ and GFAP^+^/GLAST^+^/CD133^+^/EGFR^+^ populations express the NSC transcription factor Sox2 [39] and high Sox2 levels are associated with a more proliferative state [40]. Therefore, we chose the marker combination Sox2/GFAP to capture more aNSCs. Predominantly, c-Cbl was identified in Sox2^+^/GFAP^+^ aNSCs (Figure 1E–G), Mash1^+^ TAPs (Figure 1H–J), and EGFR^+^ aNSCs/TAPs (Figure 1K–M). In brief, EGFR^+^ astrocytes represent actively dividing type B cells but niche astrocytes are negative for the EGFR [36,41]. Our results thus demonstrate c-Cbl expression in EGFR^+^ aNSCs and TAPs in the SVZ [39].

### 3.2. c-Cbl Knock-Out in TAPs Increased Proliferation In Vitro

Due to the localization of c-Cbl in EGFR-expressing cells (TAPs, C cells), the influence of a TAP-specific c-Cbl knock-out and its effects on proliferation were investigated. EGF-responsive cells of the SVZ were cultured *in vitro* in the presence of EGF to generate neurospheres. These spheres arose from cells encompassing the proliferative stages in the SVZ lineage from activated SVZ stem cells down to TAPs (C cells), with the majority of cells arising from C cells [39]. The functional impact of c-Cbl on TAPs *in vitro* was assessed in SVZ-derived neurospheres of *Cbl^fl/fl^;Cbl-b^KO^;Cbl-3^KO^;Ascl1-CreERT2* mice (*c-Cbl^ΔTAP^;Cbl-b^KO^;Cbl-3^KO^*) as compared to Cre^−^ littermates (c-*Cbl^fl/fl^;Cbl-b^KO^;Cbl-3^KO^*) (Figure 2). The functional consequences of a c-Cbl knock-out in Ascl1-positive progenitors with regard to number and size of primary neurospheres were investigated. A significant increase in the number of SVZ-derived *c-Cbl^ΔTAP^;Cbl-b^KO^;Cbl-3^KO^* neurospheres was observed after 7 d in comparison to littermate controls (275.7 ± 32.8 µm as compared to 151.7 ± 79.1). In particular, larger neurospheres (Figure 2A–D) with a size exceeding 100 μm were formed more frequently (213 ± 21.7 µm as compared to 98 ± 37.6) (Figure 2D). To address the functional role of c-Cbl in TAPs of the HC, *c-Cbl^ΔTAP^;Cbl-b^KO^;Cbl-3^KO^* mice and *Cbl^fl/fl^;Cbl-b^KO^;Cbl-3^KO^* littermates were used to isolate NSCs from the HC and they were cultured as neurospheres *in vitro*. NSCs were plated at clonal density and were cultured for 12 d. After 12 d, number and size of spheres were assessed but these parameters did not yield a significant difference in neurospheres formed from the HC (Figure 2E–H). 

These results demonstrate an increased proliferation rate of SVZ-derived but not HC-derived TAPs upon c-Cbl knock-out *in vitro* at EGF concentrations of 20 ng/mL.

### 3.3. The Loss of c-Cbl in NSCs and TAPs Does Not Affect the Proliferation and Formation of Adult-Born Neurons in the SVZ In Vivo

To determine the influence c-Cbl has on SVZ TAPs *in vivo*, TAPs and adult-born neurons were quantified in conditional *c-Cbl^ΔTAP^;Cbl-b^KO^;Cbl-3^KO^* mice as compared to *Cbl^fl/fl^;Cbl-b^KO^;Cbl-3^KO^* littermate controls. *c-Cbl^ΔTAP^;Cbl-b^KO^;Cbl-3^KO^*, a tamoxifen-inducible cell-type-specific c-Cbl knock-out system based on the expression of *Ascl1-CreERT2*, was applied, allowing for the conditional deletion of c-Cbl in TAPs. *c-Cbl^ΔTAP^;Cbl-b^KO^;Cbl-3^KO^* and *Cbl^fl/fl^;Cbl-b^KO^;Cbl-3^KO^* littermate controls received three injections of tamoxifen ip at 6 weeks of age. Mice were housed in cages with an enriched environment (EE) and were sacrificed 3 d or 28 d after administration of BrdU, an intercalating agent incorporating into the DNA of proliferating cells only, in order to quantify either early differentiation stages (NSCs, TAPs) or late stages (newborn neurons) of neurogenesis (Figure 3). Our *in vivo* studies revealed no significant reduction in NSCs (BrdU^+^/GFAP^+^/Nestin^+^) or TAPs (BrdU^+^/Mash1^+^) in *c-Cbl^ΔTAP^;Cbl-b^KO^;Cbl-3^KO^* as compared to c-*Cbl^fl/fl^;Cbl-b^KO^;Cbl-3^KO^* littermate controls (Figure 3C,F). After 28 d, no effect on the formation of adult-born neurons (BrdU^+^/NeuN^+^) was observed in the OB (Figure 3I). 

In a comparable approach, we studied the influence of c-Cbl knock-out in NSCs on the formation of adult-born neurons in the OB *in vivo* using GLAST-CreERT2 mice allowing for the deletion of c-Cbl in radial glial cells and astrocytes. The former have been shown to represent NSCs in the SVZ (Type B cells) that exhibit characteristics of astrocytes, including the expression of GFAP and GLAST [37,42]. Mice were subjected to tamoxifen injection and BrdU^+^/NeuN^+^ newborn neurons in the OB were quantified in *c-Cbl^fl/fl^;Cbl-b^KO^;Cbl-3^KO^;GLAST-CreERT2* mice (*c-Cbl^ΔNSC^;Cbl-b^KO^;Cbl-3^KO^*) as compared to *Cbl^fl/fl^;Cbl-b^KO^;Cbl-3^KO^* littermate controls. As a result, no significant effect on the number of adult-born neurons (BrdU^+^/NeuN^+^) formed in the OB of *c-Cbl^ΔNSC^;Cbl-b^KO^;Cbl-3^KO^* mice (Figure 3J–L) was observed 28 d after BrdU injection. 

In conclusion, c-Cbl knock-out in TAPs or NSCs of the SVZ *in vivo* did not cause a significant effect on adult neurogenesis *in vivo*.

### 3.4. c-Cbl Does Not Directly Affect Adult Neurogenesis in Hippocampal NSCs and TAPs 

To substantiate the lack of an influence of c-Cbl on adult hippocampal neurogenesis *in vitro* (Figure 2E–H), in an *in vivo* approach, we subjected the c*-Cbl^ΔTAP^;Cbl-b^KO^;Cbl-3^KO^* model to further analyses and quantified Ki67^+^ as well as BrdU^+^ cells in the DG of the HC. The analysis of proliferating cells (Ki67^+^) and adult-born neurons (BrdU^+^) in the DG revealed no pronounced difference in adult-born neurons in c*-Cbl^ΔTAP^;Cbl-b^KO^;Cbl-3^KO^* mice as compared to littermate controls (as measured by % of BrdU^+^/NeuN^+^ co-labeling; Figure 4A–D). Likewise, *Cbl^ΔNSC^;Cbl-b^KO^;Cbl-3^KO^* mice showed no altered number of BrdU^+^ cells in the DG granular layer 28 d after BrdU injection as compared to control mice.

Thus, data show that a c-Cbl knock-out in NSCs and TAPs of the DG in the HC did not yield a significant impact on adult neurogenesis *in vivo*.

## 4. Discussion

The SVZ is an important neurogenic niche in the adult forebrain of rodents. The activity of neural NSCs and TAPs within the SVZ is controlled by numerous extrinsic factors, whose downstream effects on the proliferation, survival, and differentiation of TAPs are mediated via intracellular signaling pathways. The present study provides new insights into the expression, localization, and physiological role of c-Cbl in the SVZ. Our results show that c-Cbl is mainly expressed in NSCs and TAPs of the subventricular niche, consistent with previous studies in which c-Cbl is a regulator of stem cell proliferation and differentiation in various tissues including the hematopoietic system, skin, and the gut [43]. We could show that c-Cbl is expressed at similar levels in NSCs, TAPs, and astrocytes within the SVZ. c-Cbl was primarily detected in EGFR-expressing activated stem and progenitor cells. This EGFR expression decreases with further differentiation towards neuroblasts [14].

Previous studies demonstrated that c-Cbl plays a critical role in regulating signaling cascades, such as receptor tyrosine kinase signaling, including the EGFR pathway, which is essential for NSC proliferation and differentiation [14]. The role of c-Cbl as a negative regulator of EGFR signaling has been well established using cancer cells or cell lines engineered to express exogenous EGFRs [20]. However, c-Cbl’s role in regulating endogenous EGFRs that contribute to tissue homeostasis is less well studied. All members of the Cbl family (c-Cbl, Cbl-b, and Cbl-3) have been reported to be E3 ligases for the EGFR [44]. 

Initially, the effects of a constitutive c-Cbl knock-out for adult neurogenesis was investigated. *In vivo*, studies of these constitutive c-Cbl knock-out mice showed no significant differences in adult neurogenesis as compared to littermate controls (data not shown). We hypothesized that any phenotypes caused may be compensated by other E3 ligases of the Cbl family. Therefore, we obtained mice from Hamid Band [45] harboring a constitutive Cbl-b as well as a Cbl-3 knock-out, allowing for the conditional deletion of c-Cbl in specific tissues. In particular, *GLAST-* and *Ascl1-CreERTs* mice have been crossed with these mice to allow for the tamoxifen-inducible deletion of c-Cbl in NSCs and TAPS, respectively.

To study the consequences for neurogenesis of a conditional c-Cbl knock-out in combination with a constitutive Cbl-b and Cbl-3 knock-out, mice were bred with *Ascl1-CreERT2* mice to allow for the specific deletion of c-Cbl in TAPs. Mice were sacrificed, brains excised, and NSCs and TAPs isolated from the SVZ to perform neurosphere assays. The cell culture medium contained 20 ng/mL EGF and FGF, a concentration triggering c-Cbl-mediated ubiquitination of the activated EGF–receptor complex [20], and thus leading to enhanced cell proliferation by RTK activation. As a result, an increased growth of spheres was observed upon c-Cbl knock-out as compared to littermate controls (floxed c-Cbl allele plus constitutive Cbl-b as well Cbl-3 knock-out). Data suggest that EGFR is negatively regulated by c-Cbl in proliferating progenitor cells. However, it is important to note that although neurosphere size provides valuable insight, it does not directly measure proliferation. 

Surprisingly, no significant role of c-Cbl for stem and progenitor cells or adult neurogenesis was observed *in vivo*. Presumably, this was due to the much lower endogenous concentrations of EGF *in vivo* as compared to EGF concentration *in vitro* (20 ng/mL). EGFR is internalized through both CME and NCE. CME is primarily coupled to EGFR recycling to the cell surface and therefore contributes to the maintenance of signaling [18]. NCE is mainly associated with lysosomal degradation of the receptor [18]. Both ubiquitination of EGFR at the plasma membrane and EGFR-NCE ubiquitination were found to be threshold dependent [20,46]. The ubiquitination threshold is mechanistically determined by the cooperative recruitment of the E3 ligase c-Cbl, in complex with Grb2, to the EGFR which is dependent on the simultaneous presence of two phosphotyrosines, pY1045 and either pY1068 or pY1086, on the same EGFR moiety. The dose–response curve of EGFR ubiquitination has been shown to correlate with the NCE mode of EGFR internalization. Sigismund et al., 2013 conclude that ubiquitination acts as a molecular switch, allowing the cell to convert a linear EGF signal into an on–off switch to activate NCE and thus regulate the fate of EGFR. The prerequisite for NCE is EGFR ubiquitination. In this context, it was shown that a triple mutant (Y1045/1068/1086F) that cannot bind to c-Cbl or Grb2 and shows greatly reduced ubiquitination showed only a moderate reduction in CME, while it was completely defective in NCE [20]. Our data suggest that the threshold for EGFR ubiquitination is not reached *in vivo* and NCE of the EGFR plays only a minor role under these conditions. While EGFR signaling is quite dominant *in vitro*, other RTKs may compensate *in vivo* and any c-Cbl phenotype may be counteracted.

Furthermore, no significant difference in the number and size of neurospheres was observed in cultures obtained from the DG of the HC. Probably, this is due to the fact that the EGFR is either not or only minimally expressed in hippocampal Ascl1-positive type 2a progenitor cells [14]. If c-Cbl is mostly involved in the degradation of this receptor, then no effects are to be expected in hippocampal neurospheres. In vivo, our data also did not show any stem or progenitor cell-specific influence of c-Cbl on adult neurogenesis in the HC. In the hippocampal SGZ, it has been reported that adult cells migrate to the GCL before differentiating into neurons. Analysis of adult-born neurons after EGF treatment showed a significant increase in BrdU-positive cells in the molecular layer. The majority of these cells were located in close proximity to the wall of the third ventricle. In the hilum and GCL, where neurogenesis normally occurs, the number of newborn cells did not change significantly after EGF treatment [47]. Taken together, we did not observe any changes in our c-Cbl knock-out lines *in vitro* or *in vivo*, suggesting that c-Cbl has no overall proliferative effect on adult neurogenesis in hippocampal progenitors.

Moreover, EGFR is only one of several possible ErbB receptor homo- and heterodimers [48]. However, the principle of c-Cbl-mediated ubiquitination cannot be applied to all ErbB receptor transfers. For ErbB3, special E3 ligases are required instead of Cbl, such as NRDP1 and NEDD4 [49]. A highly conserved Cbl-binding site is present not only in ErbB1/EGFR but also in ErbB2 and ErbB4. Replacing the ErbB1-Cbl-binding site with those of ErbB2 and ErbB4 had no effect on Cbl recruitment, receptor ubiquitination, and clearance, while retrovirus-infected NIH3T3 cells harboring the EGFR-Y1045F mutation dramatically impaired Cbl recruitment, EGFR ubiquitination, and delayed EGFR degradation [50]. These results demonstrated that the c-Cbl-binding sites of ErbB2 and ErbB4 are fully functional, and that the EGFR c-Cbl-binding site was not essential for endosomal trafficking [50]. EGFR was shown to interact directly with the CYT1 and CYT2 variants of ErbB4 and the membrane-anchored intracellular domain. The CYT2 variant, but not the CYT1 variant, protected EGFR from ligand-induced degradation by competing with EGFR for binding to a complex containing the E3 ubiquitin ligase c-Cbl and the adapter Grb2 [51]. 

In addition to its role in regulating ErbB family receptors, c-Cbl is known to control a number of other RTKs, including platelet-derived growth factor receptor (PDGFR), fibroblast growth factor receptor (FGFR), and insulin-like growth factor 1 receptor. These RTKs play critical roles in various cellular processes, influencing cell growth, migration, and other functions by fine-tuning RTK activity. For instance, PDGFR-beta, which is expressed on NPCs of the SVZ to maintain and expand the progenitor cell pool, was shown to be unaffected in terms of proliferation signaling pathways, such as Akt or Erk1/MAP, under c-Cbl and Cbl-b knock-out conditions [19,52,53]. This suggests that the regulatory role of c-Cbl may be highly specific and context dependent. In the case of the FGFR-RTK group (1–4), Cbl proteins mediate the degradation of the receptor complex through interaction with FRS2 and Grb2, thereby stopping the Akt or MAP signaling pathway [54]. In the case of the FGFR-3 subtype, Cbl does not lead to the ubiquitination of the receptor [55]. The FGFR found on GFAP-positive cells in the SVZ promotes neurogenesis primarily by activating FGF-2 ligand binding [56] and also partially mediates the effects of an enriched environment on adult hippocampal neurogenesis [57]. There are currently no data on the binding behavior of Cbl to FGFR at different ligand concentrations, however, the increased signaling of FGFR *in vitro* in the Cbl knock-out should also lead to increased proliferation of neurospheres. Thus, an explanation of the differences observed in the results by using FGFR is unlikely.

Given the diverse roles of RTKs in cellular signaling and the potential for c-Cbl to act as a key modulator of these receptors, further research is warranted to fully elucidate the spectrum of RTKs that c-Cbl targets and the context in which this regulation occurs. Such studies could provide valuable insights into the broader role of c-Cbl in neural stem cell regulation and potential therapeutic strategies for modulating RTK signaling in neurogenic niches.

In summary, c-Cbl affected NSCs and TAPs under high-EGF conditions, when EGFR signaling dominated (e.g., in neurospheres). However, under low-EGF conditions (e.g., *in vivo* or in the HC) no significant influence on these cells was observed. These observations render c-Cbl an interesting but minor player in adult neurogenesis. 

## Figures and Tables

**Figure 1 cells-12-02400-f001:**
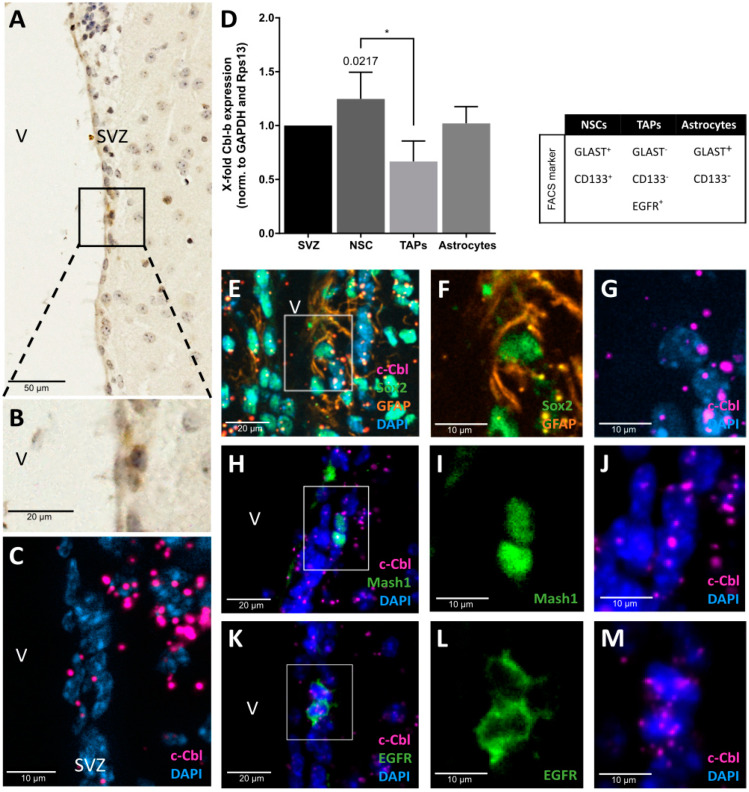
**Expression and localization of c-Cbl in the SVZ.** (**A**,**B**) IP staining of c-Cbl in the SVZ of the lateral ventricle (V). Scale bars represent 50 µm or 20 µm (magnification). (**C**) RNAscope assay confirmed the presence of c-Cbl RNA in the SVZ. Scale bar represents 10 µm. (**D**) Cells of murine SVZ were isolated by fluorescence-activated cell sorting using a combination of the following markers: GLAST^+^/CD133^+^ for NSCs, GLAST^−^/CD133^−^/EGFR^+^ for TAPs, and GLAST^+^/CD133^−^ for astrocytes. C-Cbl expression was significantly higher in NSCs as compared to TAPs (mean = 124.7% ± SD which equals 24.9%, n = 3, vs. mean = 66.7% ± SD which equals 19.1%, *p* = 0.0217). Astrocytes displayed an average c-Cbl expression in the SVZ (mean = 102% ± SD which equals 15.6%, n = 3). A combined RNAscope and IF assay revealed c-Cbl expression in Sox2^+^/GFAP^+^ NSCs (**E**–**G**), Mash1^+^ TAPs (**H**–**J**), and in EGFR^+^ cells (**K**–**M**). Data are represented as mean ± SD. * *p* ≤ 0.05.

**Figure 2 cells-12-02400-f002:**
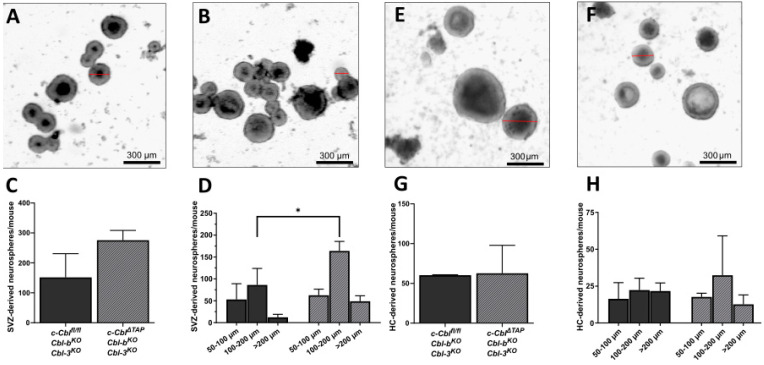
**c-Cbl knock-out affects neurosphere formation and size.** The formation of neurospheres was analyzed in (**A**) *Cbl^fl/fl^;Cbl-b^KO^;Cbl-3^KO^* control littermates and in (**B**) *c-Cbl^ΔTAP^;Cbl-b^KO^;Cbl-3^KO^* knock-out mice. (**C**) The total number of spheres derived from *c-Cbl^ΔTAP^;Cbl-b^KO^;Cbl-3^KO^* animals was increased in the neurosphere assay, but not to a significant extent (mean = 275.7 ± SD which equals 32.75 vs. mean = 151.7± SD which equals 79.10 in control mice, n = 3, *p* = 0.0662). (**D**) *c-Cbl^ΔTAP^;Cbl-b^KO^;Cbl-3^KO^* showed a significant increase in the number of neurospheres with a size between 100 and 200 μm after 7 d (mean = 164.0 ± SD which equals 21.6, n = 3 vs. mean = 86.3 ± SD which equals 37.6, n = 3 in control, *p* = 0.0037) and a slight increase in spheroids larger than 200 µm (mean = 49.0 ± SD which equals 12.5, n = 3 vs. mean = 12.3 ± SD which equals 6.8, n = 3) but no difference in the case of small neurospheres with a size of 50–100 μm (mean = 62.67 ± SD which equals 14.0, n = 3 vs. mean = 53 ± SD which equals 36.1, n = 3). Statistical analysis was performed by 2-way ANOVA analysis. Hippocampal neurosphere formation (**E**,**F**) showed neither a relevant difference between *Cbl^fl/fl^;Cbl-b^KO^;Cbl-3^KO^* control littermates and *c-Cbl^ΔTAP^;Cbl-b^KO^;Cbl-3^KO^* mice in the total number of spheres per mice (**G**) (mean = 62.7 ± SD which equals 35.2, n = 3 vs. mean = 60.0 ± SD which equals 0.6, n = 3) nor in (**H**) size. Scale bars represent 300 µm. * *p* ≤ 0.05.

**Figure 3 cells-12-02400-f003:**
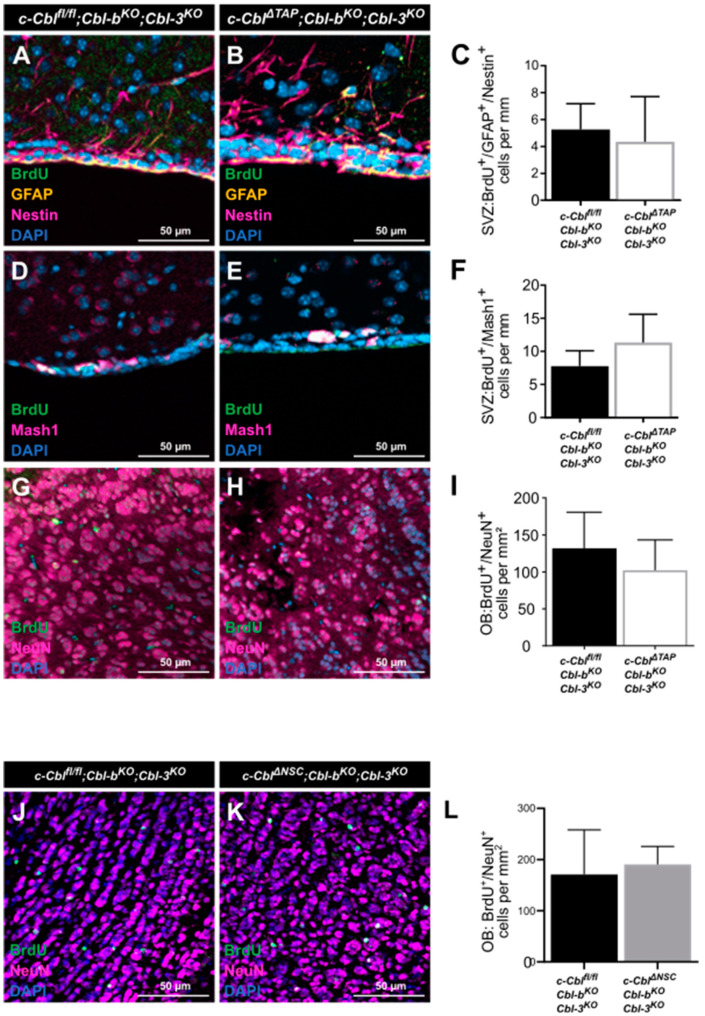
**Analysis of adult neurogenesis upon the deletion of c-Cbl in TAPs in the SVZ *in vivo*.** Markers used in IF analyses to label specific cell stages of adult neurogenesis in the SVZ of *Cbl^fl/fl^;Cbl-b^KO^;Cbl-3^KO^* control littermates, *c-Cbl^ΔTAP^;Cbl-b^KO^;Cbl-3^KO^* (**A**–**I**), or *c-Cbl^ΔNSC^;Cbl-b^KO^;Cbl-3^KO^* (**J**–**L**) knock-out mice. (**A**,**B**) The marker combination GFAP and Nestin was used to identify NSCs, (**D**,**E**) Mash1 was used for TAPs and (G,H,J,K) NeuN for mature neurons. (**A**–**C**) Quantification of BrdU^+^/GFAP^+^/Nestin^+^ cells in the SVZ of *c-Cbl^ΔTAP^;Cbl-b^KO^;Cbl-3^KO^* mice revealed no influence on the proliferation of the NSC pool (mean = 5.26 ± SD which equals 1.92 cells/mm in control mice, n = 5 as compared to mean = 4.35 ± SD which equals 3.36 cells/mm, n = 4). (**D**–**F**) Quantification of BrdU^+^/Mash1^+^ cells in the SVZ of *c-Cbl^ΔTAP^;Cbl-b^KO^;Cbl-3^KO^* showed a similar result (mean = 7.76 ± SD which equals 2.34 cells/mm in *Cbl^fl/fl^;Cbl-b^KO^;Cbl-3^KO^* control mice, n = 5 vs. mean = 11.33 ± SD which equals 4.28 cells/mm, n = 4). (**G**–**I**) After 28 d, the number of adult-born neurons in the OB was not changed in *c-Cbl^ΔTAP^;Cbl-b^KO^;Cbl-3^KO^* mice (mean = 131.9 ± SD which equals 49.31 cells/mm^2^, n = 11 vs. mean = 102.2 ± SD which equals 41.16 cells/mm^2^, n = 8) as compared to *Cbl^fl/fl^;Cbl-b^KO^;Cbl-3^KO^* littermate controls. (**J**–**L**) The number of newly differentiated adult-born neurons in the OB of *c-Cbl^ΔNSC^;Cbl-b^KO^;Cbl-3^KO^* mice showed no significant difference between knock-out and control mice (mean = 170.8 ± SD which equals 87.1 cells/mm^2^ in *Cbl^fl/fl^;Cbl-b^KO^;Cbl-3^KO^* control mice, n = 6 vs. mean = 190.5 ± SD which equals 35.2 cells/mm^2^, n = 6). Data are represented as mean ± SD; scale bars represent 50 µm.

**Figure 4 cells-12-02400-f004:**
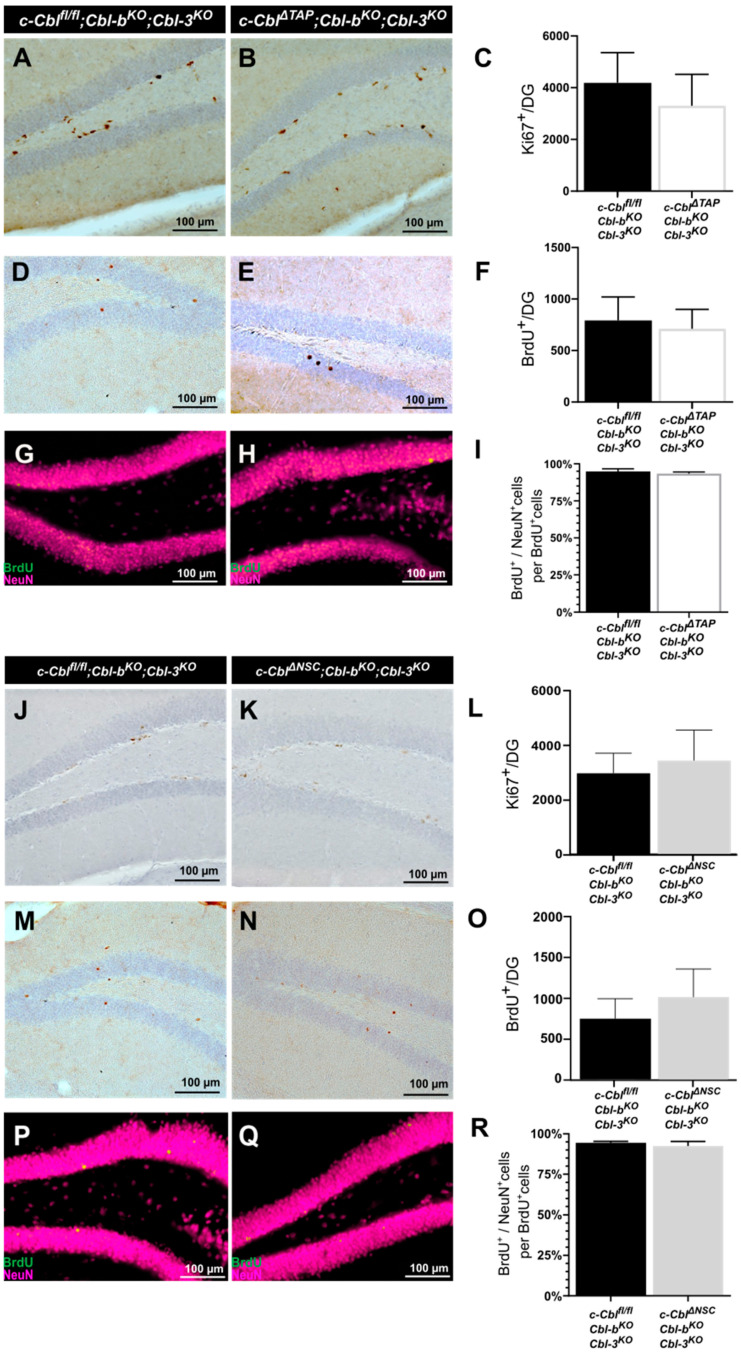
**Analysis of adult neurogenesis in the DG in mice lacking c-Cbl in TAPs and NSCs of the HC *in vivo***. Markers used in IP and IF analyses to specifically label different cell stages of adult neurogenesis in the dentate gyrus (DG). Ki67 was used as a marker for proliferating cells, NeuN for mature neurons. (**A**–**C**) Quantitative analysis of proliferating cells in the DG of *c-Cbl^ΔTAP^;Cbl-b^KO^;Cbl-3^KO^* and *Cbl^fl/fl^;Cbl-b^KO^;Cbl-3^KO^* control littermates *in vivo*. The number of Ki67^+^ cells in the DG did not significantly differ from control mice (mean = 4187.0 ± SD which equals 1172.0 cells/mm in control mice, n = 12 vs. mean = 3301.0 ± SD which equals 1215.0 cells/mm, n = 9). (**D**–**F**) The amount of BrdU^+^ adult-born cells in the DG was not affected in the *c-Cbl^ΔTAP^;Cbl-b^KO^;Cbl-3^KO^* knock-out model as compared to littermate controls (mean = 791.5 ± SD which equals 228.7 cells/mm, n = 12 in control mice vs. mean = 710.0 ± SD which equals 189.1 cells/mm, n = 9). Phenotyping of BrdU^+^ cells with a neuronal marker (NeuN). (**G**–**I**) IF showed no difference in the proportion of NeuN^+^ cells among BrdU^+^-labeled cells (mean = 94.4% ± SD which equals 1.0%, n = 3, vs. mean = 92.4% ± SD which equals 2.9%, n = 3). (**J**–**L**) Quantitative *in vivo* analysis of proliferation in the DG of *c-Cbl^ΔNSC^;Cbl-b^KO^;Cbl-3^KO^* and *Cbl^fl/fl^;Cbl-b^KO^;Cbl-3^KO^* control mice. No relevant difference in the amount of Ki67^+^ cells was detected in the DG (mean = 2880.0 ± SD which equals 717.6 cells/mm in control mice, n = 7, vs. mean = 3323.0 ± SD which equals 1083.0 cells/mm, n = 5). (**M**–**O**) The number of adult-born neurons in the DG revealed no significant difference between *c-Cbl^ΔNSC^;Cbl-b^KO^;Cbl-3^KO^* mice and *Cbl^fl/fl^;Cbl-b^KO^;Cbl-3^KO^* control littermates (mean = 740.6 ± SD which equals 244.9 cells/mm in *Cbl^fl/fl^;Cbl-b^KO^;Cbl-3^KO^* control mice, n = 7 compared to mean = 1004.0 ± SD which equals 345.4 cells/mm, n = 5). (**P**–**R**) Phenotyping of BrdU^+^ cells with a neuronal marker (NeuN). Quantification of newborn neurons (BrdU^+^/NeuN^+^) revealed a similar proportion of NeuN^+^ cells between both groups (mean = 94.8% ± SD which equals 1.8% in control mice n = 3, vs. mean = 93.4% ± SD which equals 1.1%, n = 3). Data are represented as mean ± SD; scale bars represent 100 µm.

**Table 1 cells-12-02400-t001:** Antibodies used for immunofluorescence.

Epitope	Species	Dilution	Company	Catalogue Number
BrdU	Sheep	1:400	Biozol	GTX21893
GFAP	Rabbit	1:400	DAKO	Z0334
Ki-67	Rat	1:100	Thermo Scientific	SolA15
Mash1	Rabbit	1:200	Abcam	ab211327
Nestin	Mouse	1:100	Milipore	MAB353
NeuN	Mouse	1:100	Milipore	MAB377
c-Cbl	Rabbit	1:200	Invitrogen	PA5-82992

**Table 2 cells-12-02400-t002:** PCR primer sequences.

Gene	PubMed (NM)	Forward (5′–3′)	Reverse (5′–3′)
Rps 13	NM_026533.3	ttcaccgattggctcgatac	ttatgccactagagcagagg
Gapdh	NM_001289726.1	tgaagcaggcatctgaggg	cgaaggtggaagagtgggag
c-Cbl	qMmuCID0023539, Bio-rad		
Cbl-b	qMmuCID0023240, Biorad		
Clb-3	qMmuCED0039838, Bio-rad		

## Data Availability

The data presented in this study are available on request from the corresponding author.

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
