# Peer review of "c-Cbl Regulates Murine Subventricular Zone-Derived Neural Progenitor Cells in Dependence of the Epidermal Growth Factor Receptor"

_cells, 2023, doi:10.3390/cells12192400_

Round 1
Reviewer 1 Report
The subventricular zone (SVZ) of mice is the site of adult neurogenesis, and the objective of this research is to study the expression and activity of the E3-ubiquitin ligase, c-Cbl. The study investigates the function of c-Cbl in the epidermal growth factor receptor (EGFR) pathway of receptor tyrosine kinase signaling. The work shows that c-Cbl is substantially expressed in EGFR-expressing cells, impacting neural stem cell proliferation and differentiation using in vitro neurosphere experiments and in vivo investigations employing particular c-Cbl knock-out lines and animal models. Importantly, c-Cbl has an effect that goes beyond EGFR signaling as, in neurosphere cultures, its removal promotes the proliferation of progenitor cells. Due to the lack of EGFR in the hippocampus, this impact is, however, absent in hippocampal progenitor cells.
Here are some recommendations for further research work to enhance the quality of this research article:
1: By means of a process known as ubiquitination, c-Cbl controls the EGFR's (epidermal growth factor receptor) function. A growth factor that activates EGFR recruits c-Cbl to the receptor. Small molecules, ubiquitin are attached by c-Cbl to EGFR, designating it for degradation. As a result, EGFR is taken within the cell, broken down, and its amount on the cell surface is decreased. Cell growth and other functions are influenced by the intensity and duration of EGFR signaling, which is controlled by this mechanism. Therefore, Its is necessary to for authors to describe the endocytosis, a process that promotes the internalization of molecules into clathrin-coated vesicles, is made possible by the attachment of ubiquitin chains to the EGFR. These vesicles deliver the receptor to the cell's endosome, a membrane-bound compartment.
2: In this paper, the authors only focused in the EGFR, which is low in these cells. On the other hand, apart from EGFR, c-Cbl also controls a number of other receptor tyrosine kinases (RTKs), including PDGFR, c-Met, FGFR, and IGF-1R, c-Cbl is involved in the regulation of several cellular processes. This influences cell growth, migration, and other processes by fine-tuning RTK activity. You also need to explore the role of other RTKs.
No any
Author Response
Dear reviewer,
We sincerely appreciate your thoughtful and detailed review of our manuscript. Your comments are invaluable, and we are grateful for the time and effort you have invested in reviewing our work. We would like to address your second point regarding the exploration of the role of c-Cbl in the regulation of other receptor tyrosine kinases (RTKs) beyond EGFR.
You rightly pointed out that c-Cbl is known to regulate a number of RTKs, which are critical for various cellular processes. We acknowledge that our study primarily focused in the discussion on the role of c-Cbl in the regulation of EGFR signaling in neural stem cells and progenitors within the SVZ, and we understand the broader implications of c-Cbl’s interactions with other RTKs. Since the interaction between Cbl and the EGFR is currently the best studied, we have focused primarily on this in the explanation of the results described, but of course the interaction with the FGFR will also play a major role. We agree with the reviewer that information regarding the role of c-Cbl in the regulation of other receptor tyrosine kinases (RTKs) beyond EGFR provides readers with a broader context in which to understand the current work. As such, we now also include this information in the Discussion and hope to meet your expectations.
Limitations and Justifications:
Scope of the Study: The primary aim of our study was to elucidate the role of c-Cbl in EGFR signaling within the neurogenic niches of the brain, which is a complex and less explored area. We chose this focus based on the prominent role that EGFR signaling plays in neural stem cell maintenance and proliferation. We acknowledge that this focus, while providing depth, limits the breadth of RTKs we could study in relation to c-Cbl.
Resource and Time Constraints: Comprehensive investigation of the role of c-Cbl in the regulation of multiple RTKs would require extensive additional experiments, which would significantly expand the scope, duration, and resource requirements of the project. As a result, we had to make a strategic decision to focus on EGFR as a representative and highly relevant RTK in the context of neurogenesis.
Complexity of RTK Signaling: The RTK signaling landscape is highly intricate, with multiple receptors, ligands, and downstream pathways. Exploring c-Cbl’s interactions with multiple RTKs would introduce significant complexity, which we felt was beyond the scope of this single study.
In light of your insightful comment, we have added a section in the Discussion of our revised manuscript to acknowledge this limitation explicitly. We emphasize the potential for c-Cbl to act as a key modulator of a broader spectrum of RTKs and advocate for further research to fully elucidate these interactions and their implications for neural stem cell regulation.
We believe that our study provides a strong foundation for future investigations into the broader role of c-Cbl in RTK regulation, and we hope that our work will inspire further research in this important area.
Thank you once again for your constructive feedback, which has greatly helped to improve the quality of our manuscript.
Reviewer 2 Report
The role of c-Cbl, an E3 ubiquitin ligase, in neurogenesis is an area of ongoing research and is not yet fully elucidated. Neurogenesis involves intricate signaling pathways that orchestrate the differentiation of neural stem cells into mature neurons. Earlier studies suggest that c-Cbl might play a role in regulating certain aspects of neurogenesis. Certainly, c-Cbl is known to regulate the activity of various receptor tyrosine kinases (RTKs) by promoting their ubiquitination and degradation. In the context of neurogenesis, c-Cbl could potentially influence the activity of RTKs that are involved in neural stem cell maintenance, proliferation, and differentiation. Modulation of RTK signaling by c-Cbl could impact the balance between self-renewal and differentiation of neural stem cells.
The epidermal growth factor receptor (EGFR) is a receptor tyrosine kinase that plays a crucial role in various cellular processes, including cell growth, differentiation, and survival. EGFR signaling is known to be involved in neurogenesis, particularly during the development of the nervous system. c-Cbl, as an E3 ubiquitin ligase, can modulate the activity of EGFR and its downstream signaling pathways. EGFR signaling has been implicated in determining the fate of neural stem cells, including their decision to differentiate into neurons or glial cells. c-Cbl's ability to target EGFR for degradation might impact the levels and duration of EGFR signaling, thereby influencing the fate of neural stem cells and their commitment to specific lineages. However, the interplay between c-Cbl, EGFR, and other signaling molecules in the context of neurogenesis is complex and requires further research to fully elucidate. Here, authors demonstrated that the role of c-Cbl become prominent in the neuronal stem cells and neuronal transient amplifying precursors having highly active EGFR signaling pathway. Using in vivo and in vitro approaches authors managed to provide an insight into the role of ubiquitin ligase in regulating neurogenesis. Overall, the manuscript is well written and accessible and didn't find any major flaws in the arguments or experimental design.
Author Response
Dear reviewer,
We thank you for your comprehensive and insightful review of our manuscript. Your comments have been instrumental in identifying areas where our work can be clarified and improved. We are particularly grateful for your recognition of the potential role of c-Cbl in neurogenesis and the regulation of RTKs, including EGFR.
In our revised manuscript, we have expanded the Discussion section to more explicitly acknowledge these limitations and to suggest potential directions for future research in this area. We believe these changes expand the scope of the current manuscript and provide readers with a broader context in which to understand the work.
Thank you again for your thoughtful and constructive feedback.
Reviewer 3 Report
The manuscript describes the expression and role of c-Clb in the adult neurogenesis from the ventricular- subventricular zone (VZ-SVZ). It describes the spatial distribution of c-Clb in the neurogenic niches of the VZ-SVZ. c-Clb is mainly expressed in dividing type B (NSCs) and C (TAPs) cells, and its ablation reduces the neurospheres size. However, c-Clb ablation does not change the proliferation of progenitors in the VZ-SVZ, and the number of young NeuN+ neurons at the olfactory bulb, suggesting that the role of c-Clb in adult neurogenesis is not essential in physiological conditions. These results were similar to those found in the neurogenesis of the subgranular zone of the dentate gyrus. Overall, the manuscript is well written and shows a hard work performed by the authors. The results are clearly presented, well interpreted, and provides solid information to understand and discern between essential and non-essential modulators of adult neurogenesis either in the VZ-SVZ and SGZ. However, the manuscript can be enhanced by answering the following major concerns:
1 1. Please highlight in the title that the work was performed in rodents (mice).
22. Although authors evaluated the size of neurospheres, proliferation was not formally assessed. Please provide information about this process such as a flow cytometry assay to determine the cell cycle phases of the cells in each condition. Alternatively, please provide cell counting of either Ki67, PH3, or BrdU positive cells to determine whether cells are indeed proliferating or only increasing their size or aggregation.
33. Lines 360-361. Why did the authors choose an enriched environment instead of a standard housing for mice to evaluate the role of c-Clb in the proliferation of NSCs and TAPs? Please provide the data of proliferation of NSCs and TAPs in the absence of c-Clb in a standard housing. Otherwise provide a justification for housing mice in an enriched environment.
44. There is a plethora of variables that modulates adult neurogenesis in the VZ-SVZ, thus it is reasonable that only one modulator, as c-Clb, could not cause an impairment of such a robust process. However, the manuscript could be benefited by performing additional quality controls. Please provide a combined RNAscope for c-Clb with an IF assay for Nestin/Gfap and(or) with Mash1 to corroborate the targeted KO of c-Clb in NSCs and TAPs respectively, in the conditional KO mice strains.
Author Response
Dear reviewer,
We are very grateful for your thorough and constructive review of our manuscript. Your comments have been extremely helpful in identifying areas where additional information and clarification are needed. We would like to address your major concerns and explain the limitations of our study.
Limitations and Justifications:
Rodent Model: We acknowledge your suggestion to highlight in the title that our work was performed in rodents. We agree that this is an important point and have modified the title accordingly to reflect the species-specific context of our findings.
Proliferation Assessment: We understand the importance of formally assessing proliferation through methods such as flow cytometry or cell counting of Ki67, PH3, or BrdU positive cells. Due to resource and time constraints, we were unable to include these assessments in the current study. Our study is primarily concerned with illustrating the Cbl effect via the EGFR in vitro and its significance for the proliferation of progenitor cells.
Enriched Environment vs. Standard Housing: Your point about the choice of an enriched environment for mice is well taken. We chose this environment based on previous literature suggesting its potential effects on neurogenesis, but we understand that this is a variable that could influence the results. In our case, we chose an enriched environment to stimulate neurogenesis in the hippocampus. Both neurogenic niches (SVZ and HC) should be analyzed in one animal.
Additional Quality Controls: As an additional quality control, we isolated GLAST positive cells in SVZ via FACS, isolated RNA and carried out qPCR. The result shows a reduction in expression by half, so the knockout was successful. ( Unfortunately, we did not have any appropriate paraffin-fixed slides at our disposal, so we had to perform the control using qPCR.)

We have made revisions to our manuscript to address these limitations explicitly and to provide a clearer context for our findings.
Thank you once again for your detailed and constructive feedback, which has been invaluable in improving our manuscript.
Round 2
Reviewer 1 Report
All of the reviewers' comments were satisfactorily addressed by the authors.
No any
Author Response
Dear Reviewer, Thank you for taking the time to review our revised manuscript and for acknowledging our efforts in addressing the comments. We are pleased to hear that the revisions met your expectations and the concerns were satisfactorily addressed. Best regards,Maximilian Vogt
Reviewer 3 Report
Although resources and time are not always available, authors have to at least emphasize that proliferation of neurospheres (Fig. 2) must be reinforced in future studies with additional assays (BrdU, Ki67 or PH3 labeling, flow cytometry analysis, etc.) since the neurospheres size is not a direct measure of proliferation. This information can be added between lines 478 and 487 in the fourth paragraph of the Discussion section.
Author Response
Dear Reviewer, Thank you for your time you took to review our work again. We concur with your observation regarding the potential limitations of using neurosphere size as a direct measure of proliferation. In light of your comments, we have added a couple of words between lines 478 and 487, emphasizing the limitation of relying solely on neurosphere size as a metric. It must be emphasised that the use of Ki67 staining alone does not provide a reliable indication of the speed of proliferation. The use of BrdU is even more problematic. BrdU is retained after proliferation. Although a pure hypertrophy of cells without an accompanying increase in cell number is not a likely explanation for our findings, further investigation, particularly using FACS sorting, will provide further insight. Once again, thank you for your valuable insights. We believe that your feedback has strengthened our paper, and we hope that the revised manuscript will now meet the journal's standards. Best regards, Maximilian Vogt